# Texture Evolution of a Rolled Aluminum Sheet in Multi-Pass Conventional Spinning

**Shiori Gondo *** , **Hirohiko Arai** , **Satoshi Kajino and Shizuka Nakano †**

Advanced Manufacturing Research Institute, Department of Electronics and Manufacturing, National Institute of Advanced Industrial Science and Technology (AIST), Tsukuba 305-8564, Japan; h.arai@aist.go.jp (H.A.); kajino-satoshi@aist.go.jp (S.K.); shizuka-nakano@nanograins.co.jp (S.N.)
* Correspondence: shiori-gondo@aist.go.jp; Tel.: +81-29-860-5102
† Current Address: nano grains Co. Ltd., Suwa 392-0013, Japan.

**Abstract:** This study clarified the evolution of texture in the thickness direction of the cylindrical cup which was spun from a rolled aluminum sheet in 13 passes, using electron backscatter diffraction pattern analysis. The study also obtained the relationship between the strain and layer structure, characterized by the textures. The spun workpiece had three layers in the thickness direction. The layer structure was composed of four types of textures: the Cu texture, "texture-I", which rotated 20° around <111> from the Cu texture; "texture-II", which rotated 5° around its <110> from the Cu texture; and "texture-III", which rotated 10° around its <001> from texture-I. When a blank disk had the sandwich-type layer structure Cu-I-Cu in its thickness direction, the structure changed to the Cu-II-II and Cu-III-III layer structures for the negative and positive thickness directional strains, respectively. A complex-type structure was found in the transition from Cu-I-Cu to Cu-II-II and Cu-III-III.

**Keywords:** metal spinning; aluminum; crystal orientation; texture; EBSD

## 1. Introduction

During the metal spinning process, a roller pushes the rotating material around its axis and gradually deforms it, with or without using a mandrel which has the final configuration. In particular, multi-pass conventional spinning processes have various processing conditions; for example, the roller configuration and rotational speed of the spun material (workpiece), as well as the roller paths and their parameters, are controlled. It is expected that the spinning process prevents wrinkling and fracturing, and also provides the desired configuration and mechanical properties for the workpiece, under suitable conditions. Therefore, the spinning process is one of the most efficient metal-forming processes for energy saving, high speed development, and the fabrication of high-performance materials in manufacturing.

In the past two decades, there has been rapid progress in the studies on multi-pass conventional spinning. The studies are categorized into three types based on the approaches used, namely: experimental studies, studies based on finite element method (FEM) models, and the evaluation of workpiece configuration by FEM. Experimental approaches have explored spinnability under various processing conditions, and the spinnability has been classified into three outcomes: success, wrinkling, or fracturing [1,2]. For works based on the FEM model, the meshing of the model and friction between the material and the tool were discussed. The validity of the FEM model was evaluated by comparing the height and thickness distributions of the workpiece, formed experimentally, with the results calculated by FEM [3–6]. For designs wherein the workpiece configuration was estimated by FEM,

the height and thickness of the workpiece, obtained due to the roller path, were evaluated [7–11]. Recently, stress distribution in workpieces was reported [12].

On the contrary, only a few reports have elucidated the microstructure evolution in the shear spinning process. Radović et al. investigated the crystal grain size of the spun Al-Mg alloy [13,14]. They reported an important result: the difference in the grain sizes obtained using the spinning and cold rolling processes was insignificant [14]. The results indicate that the microstructural theory on cold rolling can be adapted for spinning. Zhan et al. reported non-uniform deformation of the crystal grains along the thickness direction for hot spinning of the Ti15 alloy, depending on the differences in the friction conditions, deformation, and temperature. They also mentioned that the fiber microstructure was formed in the vicinity of the outer surface [15]. Furthermore, the following two parameters varied with the distance from the surface of the sheet in the thickness direction: (1) the angle between the rolling direction of the aluminum alloy sheet and the introduced deformation bands, and (2) the micro hardness [16]. The above results indicate the possibility of controlling the microstructure and imparting the desired mechanical properties in the thickness direction by the spinning process. Li et al. reported the configurations of the crystal grains of FeNi-based superalloys at several portions, including the corner, wall, and the open end of the workpiece that was spun via multi-pass conventional spinning [17]. Grain elongation and accumulation were observed at the thinned wall, and in the vicinity of the edge of the workpiece where the thickness increased, respectively. However, except for the previous work by Li et al. [17], few studies have been conducted that were related to the microstructure evolution in multi-pass conventional spinning.

A previous study clarified the three types of fiber textures of high-carbon steel wires that were generated by the wire drawing process, and revealed that the layer structure was composed of fiber textures at the outer and inner sides [18]. This work also determined the fact that the thinning of the fiber texture at the outer side improves the drawability and ductility of the drawn wires in the large drawing strain region. Thus, controlling the crystal orientation of the sheet in the thickness direction possibly prevents wrinkling and fracturing, and improves spinnability during multi-pass conventional spinning.

Based on the above-mentioned reports, we garnered two conclusions: a significant scope exists for further investigating the microstructure evolution in the material spun using multi-pass conventional spinning, and a possibility of controlling the crystal orientation in thickness direction does exist to prevent wrinkling and fracturing, as well as improve spinnability. Therefore, the objective of this study was to clarify the evolution of the crystal orientation, i.e., the texture distribution in the thickness direction of the spun workpiece during the multi-pass spinning process. This was aimed at obtaining one of the most fundamental microstructural changes in the multi-pass spinning process.

## 2. Materials and Methods

### 2.1. Blank Disk

A cold-rolled aluminum sheet (A1050 P-H24) was used to construct the blank disks. The thickness, inner diameter, and outer diameter of the blank disks were 1.44, 20, and 150 mm, respectively. Table 1 shows the chemical composition of the disks. The radial scratch lines with angles of −2.5 and 2.5° against the rolling direction were drawn using a scriber. The circumferential scratch curves with radii of 40, 45, 50, 55, 60, 65, and 70 mm were also drawn using the scriber. The flabellate elements surrounded by scratch lines and curves were named from layer no. 1 (LN 1) to layer no. 7 (LN 7), from the inner to outer sides in the blank disk. Figure 1a shows the positions of the scratch lines and curves on the blank disk.

**Table 1.** Chemical composition of the blank aluminum disks (mass %).

| Si | Fe | Cu | Ti | V | Al |
|------|------|------|------|------|------|
| 0.05 | 0.28 | 0.02 | 0.01 | 0.01 | Bal. |

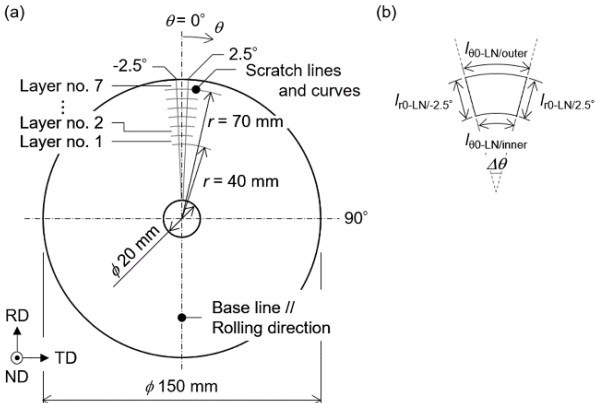

**Figure 1.** Description of parameters. (**a**) Position of the marking lines and curves on the blank disk. (**b**) Relationship among radial and circumferential directional lengths and angle between radial scratch lines.

After drawing the scratch lines and curves, the thickness $l_{t0}$, radial directional lengths $l_{r0-LN/-2.5°}$, $l_{r0-LN/2.5°}$ and angle $\Delta\theta$ between the radial scratch lines were measured using a micrometer, caliper, and protractor, respectively. The circumferential length $l_{\theta0-LN/outer,inner}$ of each element was calculated using the radius of circumferential scratch curves $r$ and angle, as expressed in Equation (1). Figure 1b shows the relationship between the radial and circumferential directional lengths and angle.

$$l_{\theta0-LN/outer,\ inner} = 2\pi r \frac{\Delta\theta}{360} \tag{1}$$

## 2.2. Spinning Process

A spinning machine constructed by Arai et al. [19] was used to spin the blank disk. The roller was equipped to the spindle axis with 45°. The roller, which was made of alloy tool steel (SKD11), had a diameter of 70 mm and a radius of roundness of 8 mm. The diameter and height of the die were 85 and 90 mm, respectively. The disk was placed between the mandrel and a jig, which was fastened to the mandrel using four bolts. A lubrication spray was applied to the surface of the disk. The roller path suggested by Sugita et al. [20] was adopted in this study. The disk was shaped into a cylindrical cup, with 13 passes at 120 rpm and 1 mm/rev.

## 2.3. Displacement Measurement

After the spinning process, the thickness $l_{tm-LN}$ of each element of the workpiece on the 1st, 5th, 9th, and 13th passes were measured using a micrometer. The subscript $m$ means pass number. The thickness directional strain $\varepsilon_{tm-LN}$ was calculated using Equation (2). The parameter $l_{t0}$ indicates the thickness of an aluminum sheet before spinning. The roller side surface of the workpiece was painted using a pencil to place its graphite into the grooves, which were carved with a scriber. The transcription was fabricated by placing the tape on the surface and peeling it off. The radial and circumferential directional length of lines $l_{rm-LN/-2.5°}$, $l_{rm-LN/2.5°}$ and curves $l_{\theta m-LN/outer}$, $l_{\theta m-LN/inner}$ which constitute elements were measured using the transcription. The radial and circumferential directional strain $\varepsilon_{rm-LN}$, $\varepsilon_{\theta m-LN}$ were calculated using Equations (3) and (4), respectively. The plastic equivalent strain, $\overline{\varepsilon_{m-LN}}$ was calculated using Equation (5).

$$\varepsilon_{tm-LN} = \frac{l_{tm-LN}}{l_{t0}} \tag{2}$$

$$\varepsilon_{rm-LN} = \frac{l_{rm-LN/-2.5°} + l_{rm-LN/2.5°}}{l_{r0-LN/-2.5°} + l_{r0-LN/2.5°}} \tag{3}$$

$$\varepsilon_{\theta m-LN} = \frac{l_{\theta m-LN/outer} + l_{\theta m-LN/inner}}{l_{\theta 0-LN/outer} + l_{\theta 0-LN/inner}} \tag{4}$$

$$\overline{\varepsilon_{m-LN}} = \sqrt{\frac{2}{3}\left(\varepsilon_{rm-LN}{}^2 + \varepsilon_{\theta m-LN}{}^2 + \varepsilon_{tm-LN}{}^2\right)} \tag{5}$$

*2.4. EBSD Analysis*

The workpiece was cut using a fine cutter along the scratch lines and curves. For the elements at layers 4 and 7 on the 1st, 5th, and 13th passes, and the elements at layers 2, 4, 6, and 7 on the 9th pass, the pieces were embedded using an electrically conductive resin. Then, the sample was polished in order for the vertical plane of the transversal direction to become the observation plane.

After polishing the plane, it was observed through the field emission-scanning electron microscope (FE-SEM) at 400× magnification, using the shifting observation visual field in the thickness direction. The Kikuchi patterns of Al were obtained via the backscatter diffraction pattern (EBSD) analysis at each observation visual field. After obtaining the Kikuchi patterns, inverse pole figure (IPF) maps were drawn using a software (Bruker Co., Esprit 2.1.0, Berlin, Germany). Furthermore, the orientation distribution function (ODF) was drawn every one-third of the observation visual field.

## 3. Results

*3.1. Thickness Directional and Equivalent Plastic Strains*

Figure 2 shows the thickness directional and plastic equivalent strains at layers 4 and 7 on the 1st, 5th, and 13th passes, and layers 2, 4, 6, and 7 on the 9th pass. The plastic equivalent strain increased with increasing pass number. The thickness directional strain of layers 2, 4, and 6 decreased with increasing pass number, while the thickness directional strain at layer 7 increased.

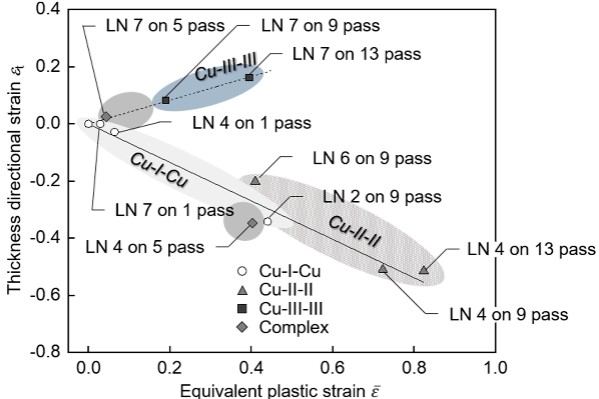

**Figure 2.** Relationship between the thickness directional and equivalent plastic strains of the spun aluminum disk. Four types of plots: circle, triangle, square, and diamond indicate layer structures characterized by textures: Cu-I-Cu, Cu-II-II, Cu-III-III, and complex types, respectively. The Cu texture was {112}<111>. Texture-I rotated 20° around its <111> from the Cu texture. Texture-II rotated 5° around its <110> from the Cu texture. Texture-III rotated 10° around its <001> from texture-I.

*3.2. Crystal Orientation*

Figure 3 shows the scanning electron microscope (SEM) images and IPF maps in the thickness direction. The SEM images and IPF maps were shown to overlap the parts. The IPF X map shows the direction of the crystal orientation, which was parallel to the rolling direction (RD). The IPF Y map shows the direction of the crystal orientation, which was parallel to the normal direction (ND). The IPF Z map shows the direction of crystal orientation, which was parallel to the transversal direction (TD). The IPF maps show slightly dark regions along the upper and lower sides of each

analysis area. There were difficulties analyzing these dark areas, and some plots were detected as a zero solution. These difficulties were due to two reasons: (1) the sample was analyzed at a low magnification, to clarify the crystal orientation distribution in the thickness direction, and (2) there were difficulties detecting the Kikuchi patterns of samples with a high strain. The analysis and sample conditions caused the small dark areas on the IPF maps.

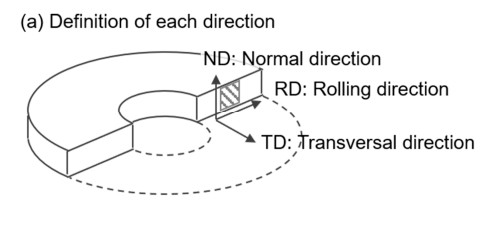

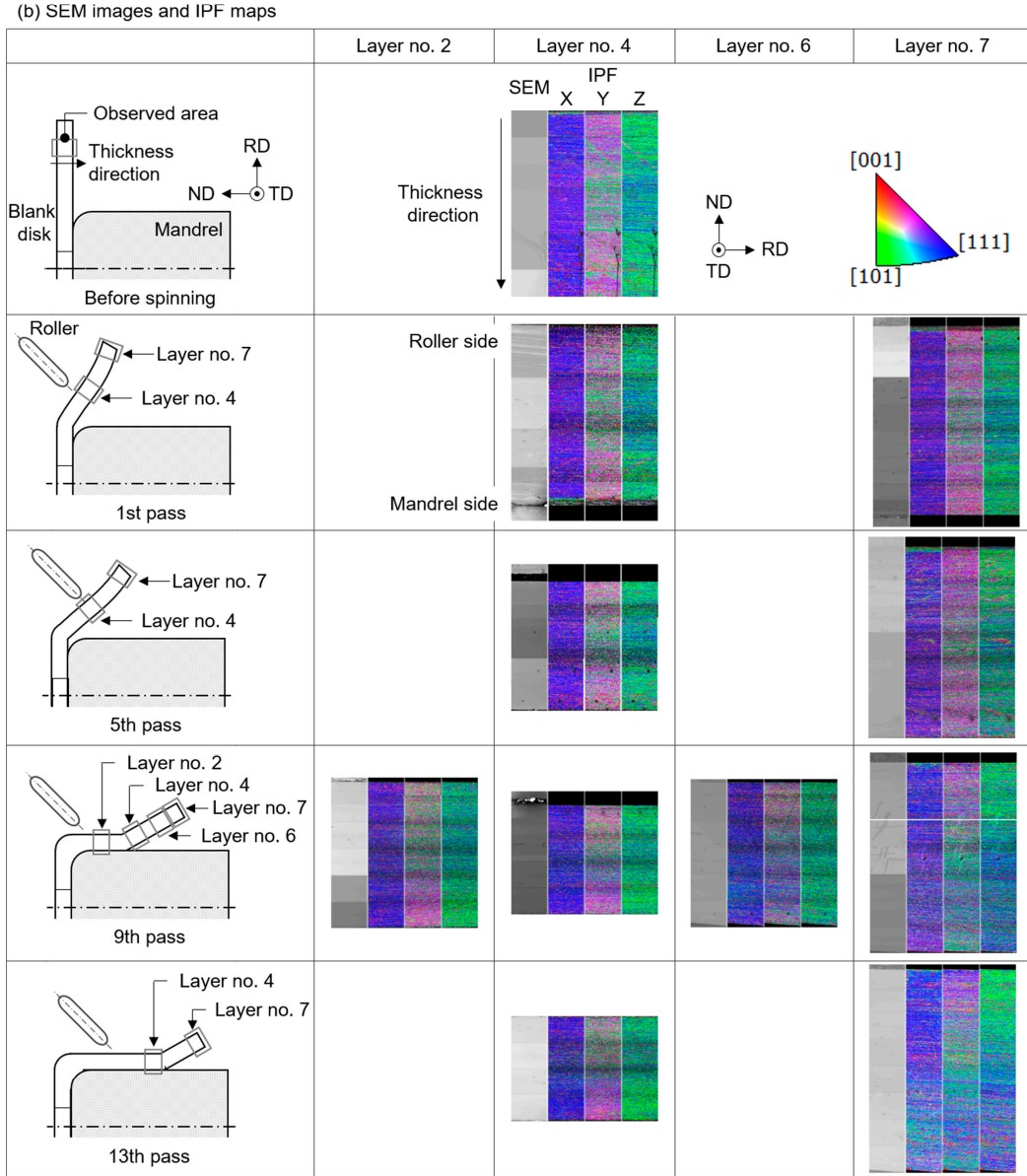

**Figure 3.** Evolution of the crystal orientation distribution in the thickness direction with spinning process. (**a**) Definition of the directions in the blank disk: rolling (RD), transversal (TD), and normal directions (ND). (**b**) Scanning electron microscope (SEM) images and inverse pole figure (IPF) maps for the blank disk and layers on the 1st, 5th, 9th, and 13th passes.

In the IPF Y and Z maps of the blank disk, identical colors were approximately one third in thickness. The colors of the outer and inner sides were different. Similar colors were shown in the entire IPF X map. The IPF maps of the workpiece in the early passes were similar to those of the blank disk. In the later passes, the colors of the IPF maps at the mandrel side of the workpiece were different from those of the blank disk. Figure 4 shows the ODF at the second and tenth from the roller side for the layer 7 on the ninth pass as examples of the ODF. The peak positions varied with the analysis fields. The high-intensity crystal orientation obtained from the ODF did not change with an increase in the number of plots with a zero solution.

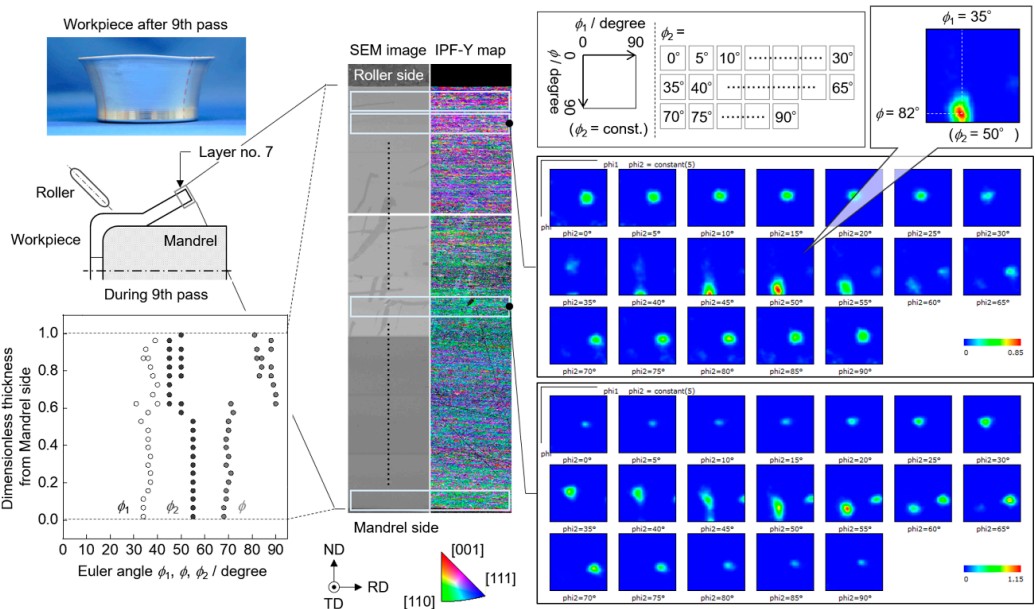

**Figure 4.** Photograph of the workpiece after the 9th pass, orientation distribution function (ODF) obtained at local areas for layer 7 on the 9th pass, and SEM images, IPF Y maps, and distribution of the Euler angle at a strong peak of the ODF in the thickness direction.

## 4. Discussion

### 4.1. Crystal Orientaiton Distribution

The crystal orientations observed in the blank disk implied a rolling texture formed by the rolling process, because the IPF X map of the blank disk indicated <111>. The IPF maps in Figure 3 indicate the change in the rolling texture of a rolled aluminum sheet by the spinning process. Figure 4 shows one part of the ODF ($\phi_2$ = 50°), which was obtained at the second analysis field from the roller side in layer 7 on the ninth pass. A strong peak was detected at the Bunge Euler angles ($\phi_1$, $\phi$, $\phi_2$) = (35, 82, 50). The Euler angles ($\phi_1$, $\phi$, $\phi_2$) at a strong peak for other analysis fields were measured. The distributions in the thickness direction of the Euler angles are shown in Figure 4. Figure 5 shows the distribution in the thickness direction of the Euler angles for each layer and pass.

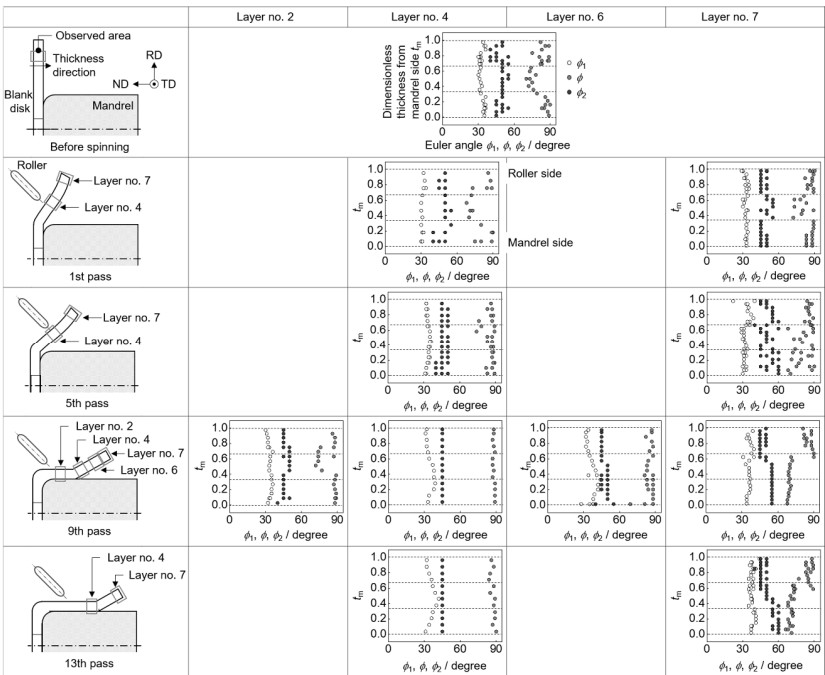

**Figure 5.** Evolution of the distribution of the Euler angle at the strong peak of ODF in the thickness direction with spinning process.

For the blank disk, the Euler angles at the outer one third in thickness were approximately $(\phi_1, \phi, \phi_2) = (35, 90, 45)$. The Euler angles at the inner one third in thickness were approximately $(\phi_1, \phi, \phi_2) = (35, 70, 45)$. The blank disk had a sandwich-type layer structure composed of two types of textures: the Euler angles on the outer $(\phi_1, \phi, \phi_2) = (35, 90, 45)$ and inner sides $(\phi_1, \phi, \phi_2) = (35, 70, 45)$. The distributions of the Euler angles in the thickness direction at layers 4 and 7 on the first pass, and layer 2 on the ninth pass were similar to those of the blank disk.

The Euler angles $(\phi_1, \phi, \phi_2)$ were (35, 90, 45) one third in thickness from the roller side at layer 4 on the 9th and 13th passes, and layer 6 on the 9th pass. They were the same as that of the blank disk. On the contrary, the Euler angles were $(\phi_1, \phi, \phi_2) = (35, 70, 55)$ two thirds in thickness from the mandrel side. The Euler angles $(\phi_1, \phi, \phi_2)$ were (35, 90, 45) one third in thickness from the roller side at layer 7 on the 9th and 13th passes. They were also the same as that of the blank disk. On the contrary, the Euler angles were $(\phi_1, \phi, \phi_2) = (40, 90, 45)$, two thirds in thickness from the mandrel side. Hence, the spun workpiece remained a layer structure in the thickness direction. The texture in the two thirds in thickness from the mandrel side was changed by the spinning process. For the fifth pass, the mixed textures were detected.

The rolling texture on the outer side of the blank disk corresponded to the Cu texture. The Euler angles in this study were different from those in [21] because of the different sample coordinate systems. The texture $(\phi_1, \phi, \phi_2) = (35, 70, 45)$ on the inner side corresponded to the texture that rotated 20° around its <111> from the Cu texture. The unit lattices in Figure 6 were drawn using a software (Kobe University, ReciPro$^{TM}$). The texture $(\phi_1, \phi, \phi_2) = (40, 90, 45)$ in two thirds from the mandrel side thickness at layer 4 on the 9th and 13th passes, and layer 6 on the 9th pass, corresponded to the texture that rotated 5° around its <110> from the Cu texture. The texture $(\phi_1, \phi, \phi_2) = (35, 70, 55)$ in two thirds from the mandrel side thickness at layer 7 on the 9th and 13th passes corresponded to the texture that rotated 20° around its <111> from the Cu texture, and 10° around its <001>.

Each texture was defined as follows: The texture that rotated 20° around its <111> from the Cu texture was defined as "texture-I"; the texture that rotated 5° around its <110> from the Cu texture was defined as "texture-II"; and the texture that rotated 10° around its <001> from texture-I was defined as "texture-III". As shown in Figure 7, the transition of the layer structure with spinning can

be explained as follows. In the entire region in the early passes, and where the roller did not touch the workpiece in the later passes, the workpiece had a sandwich-type layer structure, similar to the blank disk. The structure was Cu-I-Cu from the roller side. In contrast, when the roller touched the workpiece in the later passes, the workpiece had a Cu-II-II layer structure in the region excluding the vicinity of the edge, and the Cu-III-III layer structure in the vicinity of the edge, respectively. In the middle passes, the workpiece had mixed structures of Cu-I-Cu, Cu-II-II, and Cu-III-III.

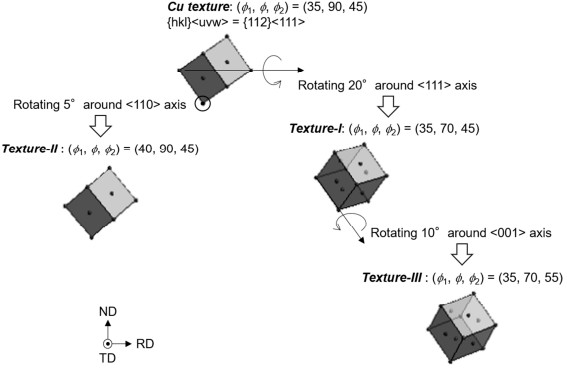

**Figure 6.** Relationships among the stable directions: Cu texture, texture-I, texture-II, and texture-III.

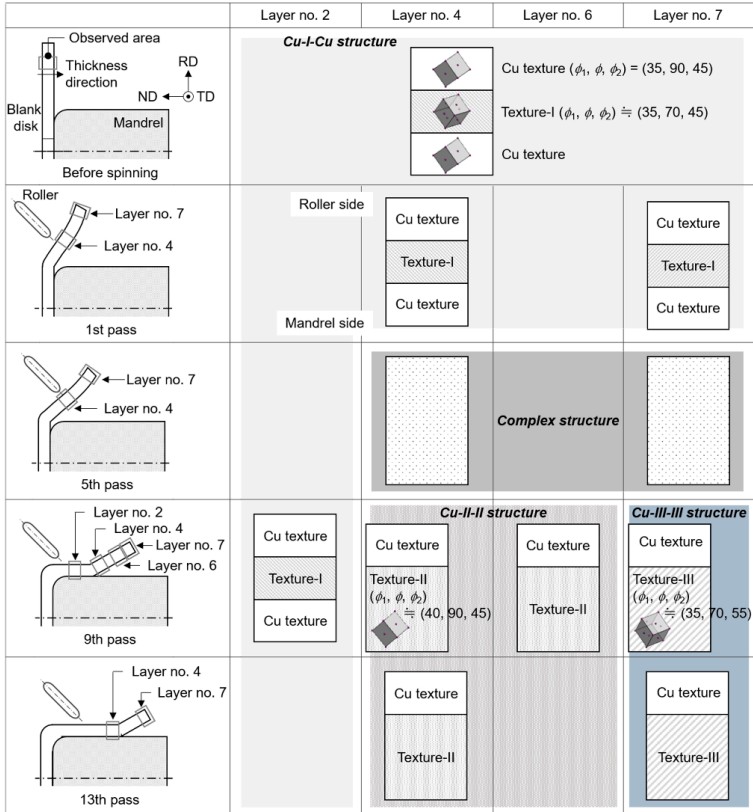

**Figure 7.** Evolution of the layer structure characterized by the textures with spinning process. Four types of layer structures were obtained by the conventional spinning process: Cu-I-Cu, Cu-II-II, Cu-III-III, and complex types. The Cu texture was {112}<111>. Texture-I rotated 20° around its <111> from the Cu texture. Texture-II rotated 5° around its <110> from the Cu texture. Texture-III rotated 10° around its <001> from fiber texture-I.

## 4.2. Relationship between the Layer Structure and Strain

The plots in Figure 2 are categorized by their layer structures. The Cu-I-Cu layer structure was found in the blank disk and in the workpieces with small plastic equivalent strain. The Cu-II-II layer structure was found in the workpieces with large plastic equivalent and negative thickness directional strains. The Cu-III-III layer structure was found in the workpieces with positive thickness directional strain. The complex-type structure was found in the transition from Cu-I-Cu to Cu-II-II or Cu-III-III.

## 5. Conclusions

This study clarified the evolution of layer structure, characterized by textures in the thickness direction of a rolled aluminum sheet which was spun in the multi-pass spinning process, using the electron backscatter diffraction pattern analysis. The important results obtained are as follows:

1.  The crystal orientation of the workpiece changed two thirds in thickness from the mandrel side by the spinning process.
2.  For the rolled aluminum sheet, the workpiece had four types of texture: the Cu texture, "texture-I", which rotated 20° around <111> from the Cu texture; "texture-II", which rotated 5° around its <110> from the Cu texture; and "texture-III", which rotated 10° around its <001> from the texture-I.
3.  When a blank disk had the sandwich-type layer structure Cu-I-Cu in the thickness direction, the structure changed to Cu-II-II and Cu-III-III layer structures for the negative thickness and positive thickness directional strains, respectively. The complex-type structure was found in the transition from Cu-I-Cu to Cu-II-II and Cu-III-III.

**Author Contributions:** Conceptualization, S.G., H.A. and S.N.; Data curation, S.G.; Formal analysis, S.G.; Investigation, S.G. and H.A.; Methodology, S.G. and H.A.; Project administration, S.G.; Software, H.A.; Validation, S.G.; Visualization, S.G.; Writing-original draft, S.G.; Writing-review & editing, S.G., H.A., S.K. and S.N. All authors have read and agreed to the published version of the manuscript.

**Funding:** This research was partially supported by SUZUKI FOUNDATION.

**Conflicts of Interest:** The authors declare no conflict of interest.

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
