# Peer review of "Texture Evolution of a Rolled Aluminum Sheet in Multi-Pass Conventional Spinning"

_metals, doi:10.3390/met10060793_

Round 1

Reviewer 1 Report

The article shows the evolution of the crystal orientation in the thickness direction of the spun workpiece, during the multi-pass spinning process. The manuscript could be considered for publication after minor revision. Some points require changes: 1) Images in Figure 3 are very small. They should be submitted with a higher resolution 2) The description about IPF maps could be improved. There are several darker areas in the IPF maps following the thickness direction. What are? 3) Conclusions should be improved and better describe that results could be improve spinnability and prevent fracturing.

Reviewer 2 Report

In this paper, the authors describe the texture evolution in an Al sheet during multipass spinning. I find the objective, experimental approach results and discussions are sufficient and recommend for publication in Metals. Among the minor issues, the authors may revisit the scientific language and make corrections as required.

Reviewer 3 Report

The manuscript entitled: 'Texture evolution of a rolled aluminum sheet in multi-pass conventional spinning' focuses on the evolution of texture in thermomechanically processed Al sheet. I have the following concerns with the manuscript.

The authors have defined the presence of four different types of texture with the rolling of the Al-sheet. However, they have failed to explain the origin of these four different types of texture connecting with the thermomechanical process and the degree of strain introduced in the material, which is vital to understand the evolution of texture in these processes.

In addition, authors may introduce the XRD patterns of the sheets before and after rolling and calculate the internal strain in the material. It may be very useful to correlate with the texture.

HR-SEM microstructure of the sample before and after rolling should be introduced. 

Round 2

Reviewer 3 Report

The authors have partially answered the queries raised and hence I cannot recommend the manuscript for publication in the present state. 

I would still urge the authors to introduce the XRD patterns of the sheets before and after rolling and calculate the internal strain in the material. It may be very useful to correlate with the texture. HR-SEM microstructure of the sample before and after rolling should be introduced. Without the metallurgical understanding of the material just the texture evolution do not add any value and do not take the science forward.

Round 3

Reviewer 3 Report

I would partially agree with the response from the authors. However, XRD might be very helpful to evaluate the internal strain in the material. I reluctantly recommend for publication of the manuscript in the present form.